# Robust Natural Language Representation Learning for Natural Language Inference by Projecting Superficial Words Out

## Abstract

In natural language inference, the semantics of some words do not affect the inference. Such information is considered superficial and brings overfitting. How can we represent and discard such superficial information? In this paper, we use first order logic (FOL) - a classic technique from meaning representation language - to explain what information is superficial for a given sentence pair. Such explanation also suggests two inductive biases according to its properties. We proposed a neural network-based approach that utilizes the two inductive biases. We obtain substantial improvements over extensive experiments.

## 1 Introduction

In natural language inference (Bowman et al., 2015), the semantics of some words do not affect the inference. In figure 1a, if we discard the semantics of some words (e.g. *Avatar*, *fun*, *adults*, *children*) from $s_1$ and $s_2$, we obtain $s_1'$ and $s_2'$, respectively. Without figuring out the specific meaning of these words, one can still infer that they are contradictory. In this case, the semantics of *Avatar*, *fun*, *adults*, and *children* are superficial for the inference.

Such superficial information brings overfitting to models. Recent studies already noticed that superficial information will hurt the generalization of the model (Jia and Liang, 2017), especially in unseen domains (Wang et al., 2019). Without distinguishing the superficial semantics, an NLI model can learn to predict *contradiction* for sentence pairs with "children" or "adults" by example 1 in Figure 1a. On the other hand, if we discard the superficial information during inference, we can prevent such overfitting.

---

$s_1$: *Avatar is fun for children, not adults.*  $s_2$: *Avatar is fun for adults, not children.*
**Label:** contradiction

After discarding *Avatar*, *fun*, *adults*, *children* :
$s_1'$: *A is B for C, not D.*  $s_2'$: *A is B for D, not C.*
**Label:** contradiction

After discarding *Avatar*, *fun*, *adults*, *children* and their correspondence information:
$s_1''$: *— is — for —, not —.*  $s_2''$: *— is — for —, not —.*
**Label:** unknown

---

(a)

---

$s_3$: *Avatar is fun for all people.*  $s_4$: *Avatar is fun for adults only.*
**Common sense:** *People include adults and children.*
**Label:** contradiction

---

(b)

Figure 1: Examples.

Some approaches have been proposed to reduce such overfitting. HEX (Wang et al., 2019) identifies the superficial information by projecting the textural information out. HEX defines the textural information w.r.t. the background of images for image classification, which cannot be generalized to other tasks (e.g. NLP). For NLP, the attention mechanism (Bahdanau et al., 2015) is able to discard some words by assigning them low attention scores. But such mechanism is more about the semantic similarity or relatedness of the words, not the superficial semantics. In example 1 of figure 1, the two *Avatar* in the two sentences will have a high attention score, since their similarity is 1 (Vaswani et al., 2017). But we have shown that these words are superficial for inference. So previous approaches cannot be applied to modeling the superficial information in natural language inference.

On top of that, a more critical issue is *the lack of mathematical definition of such superficial information* in previous studies. Why do people think the semantics of *adults* and *children* are superficial? In this paper, we tackle this question via the toolkit of first-order logic (FOL). FOL is a classic technique of meaning representation language, which provides a sound computational basis for the inference. We explain such superficial information from the perspective of FOL. Furthermore, such explanation suggests two inductive biases, which are used to design our NLI model.

| | |
|---|---|
| $FOL(s_1)$: $\forall x, Fun(x, Avatar) \Rightarrow Adult(x) \wedge \neg Child(x)$ 
 $FOL(s_2)$: $\forall x, Fun(x, Avatar) \Rightarrow Child(x) \wedge \neg Adult(x)$ 
 **Label:** contradiction | $FOL(s_3)$: $\forall x, People(x) \Rightarrow Fun(x, Avatar)$ 
 $FOL(s_4)$: $\forall x, Fun(x, Avatar) \Rightarrow Adult(x)$ 
 $FOL(CS)$: $\exists x, People(x) \wedge \neg Adult(x)$ 
 **Label:** contradiction |
| (a) | (b) |

Figure 2: The FOLs of figure 1.

By representing natural language sentences by FOL, the sentence pair and its FOLs are logically equivalent. The conversion of figure 1a is shown in figure 2a. The entailment (resp. contradiction) between $s_1$ and $s_2$ is equivalent to $FOL(s_1) \models FOL(s_2)$ (resp. $FOL(s_1) \models \neg FOL(s_2)$). Thus we successfully convert the problem of identifying superficial information in NLI to identifying the superficial information in FOL inference.

The superficial information exists in the non-logical symbols in FOL. From the specification of the FOL representation (Russell and Norvig, 1995), the symbols of FOL include the logical symbols and non-logical symbols. In figure 1a, the contradiction remains if we discard the semantics of *Avatar*, *fun*, *adults*, *children*, which are non-logical symbols. We can surely change these non-logical symbols to new symbols without changing the results of $FOL(s_1) \models FOL(s_2)$ or $FOL(s_1) \models \neg FOL(s_2)$.

However, there is a big gap between the FOL representation and the natural language: people use common sense when understanding the natural language. For example, people are able to infer the contradiction between $s_3$ and $s_4$ in figure 1b, because they have the common sense that people include adults and children. The FOLs of $s_3$, $s_4$ and the common sense are shown in figure 2b. With the common sense, the contradiction between $s_3$ and $s_4$ is equivalent to $CS \wedge FOL(s_3) \models \neg FOL(s_4)$, where $CS$ denotes the FOL of the common sense.

With the common sense, some non-logical symbols in the two sentences are not superficial, because we need these non-logical symbols for joint inference with the common sense. For example, in figure 2b, the non-logical symbols $Adult$ and $People$ are not superficial. This brings the major challenge of using FOL to identify the superficial information, because the common sense can hardly be obtained.

Since the common sense is unknown, we restrict the definition of superficial symbols. We regard a non-logical symbol as superficial, if it is superficial for all possible common sense. We show the necessary condition of the superficial symbols to avoid the effect of the common sense, which is unknown. We show that the necessary condition is related to the semantical formula-variable (FV) independence (Lang et al., 2003), which is NP-complete. Nevertheless, the properties of the FOL suggest two inductive biases for superficial information identification: word information discard

and correspondence information representation. We propose a neural network-based approach to incorporate such two inductive biases.

We point out that we need to retain the correspondence information of the discarded words. From the perspective of FOL, although the semantics of some non-logical symbols are independent for inference, the correspondence information still affects the inference. More specifically, we need to represent the occurrence of one word in different positions in the sentence pair. This is also intuitive from the perspective of natural language inference. For example, in figure 1a, although *adults* and *children* are superficial, we need to be aware that *for* is followed by *adults* in $s_1$, while *for* is followed by *adults* in $s_2$. Otherwise, as illustrated in $s_1''$ and $s_2''$, we cannot infer their relation.

We summarize our contributions in this paper below:

- We proposed the problem of identifying and discarding superficial information for robust natural language inference. We use FOL to precisely define what information is superficial.

- We analyze the superficial information from the perspective of FOL. We show that the superficial non-logical symbols are related to the semantical formula-variable (FV) independence in reasoning. We give two properties of the superficial information, and design neural networks to reflect the two inductive biases accordingly.

- We implement a neural network-based algorithm based on the two inductive biases. The experimental results over extensive settings verify the effectiveness of our proposed method.

## 2 RELATED WORK

**Learning Robust Natural Language Representation.** Noticing that traditional neural networks for the natural language easily fail in adversarial examples (Jia and Liang, 2017; Rajpurkar et al., 2018), learning robust representations is important for NLP tasks. A critical metric of the robustness is whether the model can be applied to a different data distribution Wang et al. (2019). Adversarial training (Goodfellow et al., 2014) is one way to increase the robustness for NLP models (Goodfellow et al., 2014). It has been applied to NLP tasks such as relation extraction (Wu et al., 2017), sentence classification (Liu et al., 2017). The idea is to use adversarial training to learn a unified data distribution for different domains. But the domain-specific information of the target domain must be known. In contrast, we want to learn a robust model that can be applied without knowing the target domain. And we learn robust representations by projecting superficial information out. HEX (Wang et al., 2019) is a recent approach to project textural information out of images. It relies on two models to represent the whole semantics and superficial semantics, respectively. Few studies reveal how to do this for NLP.

**Omit Superficial Information by Attention.** The attention mechanism (Bahdanau et al., 2015) gives different weights to different words according to their attention scores. Attention and its variations are successful in many NLP tasks (Vaswani et al., 2017; Devlin et al., 2018; Cui et al., 2019). Literally, attention also projects some words out by assigning them low attention scores. However, the attention scores cannot be used to project superficial information of the overlapping words out. Attention gives two words high attention scores if they are similar or equal, even if they are superficial. So we cannot use attention to discard superficial information of overlapping words. As illustrated in section 1, much superficial information for cross-sentence inference lies in these overlapping words.

**Natural Language Inference** uses neural networks to improve its accuracy (Bowman et al., 2016). Recent studies (Shen et al., 2018b;a) apply attention mechanism (Bahdanau et al., 2015) to model the word correlations. State-of-the-art approaches (Devlin et al., 2018; Liu et al., 2019) are fine-tuned over the large-scale pre-training models.

## 3 PRELIMINARIES OF FIRST-ORDER LOGIC

According to the specification of FOL in (Russell and Norvig, 1995), the atoms of FOL include logical symbols (connective, quantifier), and non-logical symbols (constant, variable, predicate, and function). We show the context-free grammar specification of the syntax of them in Table 6. We

omit the syntaxes of more complicated elements of FOL (e.g. formula) since they are irrelevant to this paper. Examples of FOLs are shown in figure 2.

# 4 PROBLEM ANALYSIS: FROM THE FIRST-ORDER LOGIC PERSPECTIVE

## 4.1 FROM NATURAL LANGUAGE INFERENCE TO FIRST-ORDER LOGIC INFERENCE

Firstly, we revealed the relation between natural language inference and FOL inference. The general purpose of NLI is to determine the *contradiction*, *entailment*, and *neutral* relations of two sentences. If we convert the two sentences into two FOLs, the relation of the FOLs directly reflects the inference label of the two sentences, as shown in Table 1.

| NLI label | FOL | FOL with common sense |
|---|---|---|
| entailment | $FOL(s_1) \models FOL(s_2)$ | $CS \wedge FOL(s_1) \models FOL(s_2)$ |
| contradiction | $FOL(s_2) \models \neg FOL(s_1)$ | $CS \wedge FOL(s_1) \models \neg FOL(s_2)$ |
| neural | otherwise | otherwise |

Table 1: NLI labels and FOL relations.

People understand natural language with external common sense. We show the mapping between natural language inference and FOL inference with common sense in table 1.

Obviously, the conversion from a natural language sentence to a FOL sentence is not trivial. We highlight that our paper do not require an algorithm to implement such conversion. We only use FOL to explain the superficial information in NLI, and to suggest inductive biases for our algorithm.

## 4.2 SUPERFICIAL INFORMATION ANALYSIS IN FOLs

We analyze the superficial information in the entailment relation. The other two relations (i.e. contradiction and neural) can be analyzed similarly. Note that the entailment relation depends on the common sense, which is unknown for NLI. So we restrict the definition of the superficial information in FOLs w.r.t. all possible common sense.

**Definition 1.** *Given $FOL(s_1)$, $FOL(s_2)$, with non-logical symbol space $V$, we define a non-logical symbol $ns \in V$ is superficial, if replacing $ns$ to with $ns'$ (s.t. $ns' \notin V$) in $FOL(s_1)$, $FOL(s_2)$ satisfies that $\forall CS$,*

$$CS \wedge FOL(s_1) \models FOL(s_2) \tag{1}$$

*is equivalent to*

$$CS \wedge FOL'(s_1) \models FOL'(s_2) \tag{2}$$

*, where $FOL'(s_1)$, $FOL'(s_2)$ are the FOLs after the replacement.*

Since $CS$ can have arbitrary sentences, analyzing the superficial symbols with $CS$ is challenging. We first derive a necessary condition in theorem 1 to avoid the effect of $CS$.

**Theorem 1.** *Given $FOL(s_1)$, $FOL(s_2)$, a non-logical symbol $ns$ is superficial, only if*

$$FOL(s_1) \models FOL(s_2) \tag{3}$$

*is equivalent to*

$$FOL'(s_1) \models FOL'(s_2) \tag{4}$$

Theorem 1 provides a necessary condition for identifying superficial non-logical symbols that only considers $FOL(s_1)$ and $FOL(s_2)$. Thus it is feasible to address whether the necessary condition is true by only using $FOL(s_1)$ and $FOL(s_2)$. The condition in theorem1 is similar to the semantic FV independence problem (Lang et al., 2003) in reasoning, which is NP-complete (Lang et al., 2003). However, we can still utilize its properties to help identify the superficial information. We show this in theorem 2.

**Theorem 2.** *Given two FOLs $FOL^{\mathbb{A}}(s_1)$ and $FOL^{\mathbb{A}}(s_2)$, with their non-logical symbol set $\mathbb{A} = \{a_1, \cdots, a_n\}$. $\forall \mathbb{B} = \{b_1, \cdots, b_n\}$, where each $b_i$ is a non-logical symbol, if we replace each $a_i$ with $b_i$ in $FOL^{\mathbb{A}}(s_1)$ and $FOL^{\mathbb{A}}(s_2)$ to get $FOL^{\mathbb{B}}(s_1)$ and $FOL^{\mathbb{B}}(s_2)$ respectively, we have*

$$FOL^{\mathbb{A}}(s_1) \models FOL^{\mathbb{A}}(s_2) \tag{5}$$

*is equivalent to*

$$FOL^{\mathbb{B}}(s_1) \models FOL^{\mathbb{B}}(s_2) \tag{6}$$

. *Note that both $\mathbb{A}$ and $\mathbb{B}$ contain $n$ distinct non-logical symbols.*

Theorem 2 points out that, from the perspective of FOL, the semantics about non-logical symbols do not affect the implication of two FOLs. Note that we need to guarantee that the $n$ non-logical symbols in $\mathbb{B}$ are distinct. We need to reserve the correspondence of these symbols to reserve their relation. The theorem is easy to prove because uniformly modifying the non-logical symbols in two FOL does not change their implication.

### 4.3 FROM SUPERFICIAL INFORMATION IN FOLS TO INDUCTIVE BIAS IN NEURAL NETWORKS

The properties of superficial information in FOLs suggests what information should be discarded in natural language inference. In this subsection, we elaborate two types of inductive biases, and how we use neural network to represent these inductive biases. More details of the neural network are shown in section 4.4.

**Word Information Discard** From theorem 1, the necessary condition of a word being superficial is that it corresponds to a non-logical symbol, and $FOL(s_1) \models FOL(s_2)$ is equivalent to $FOL'(s_1) \models FOL'(s_2)$. As we use the word embedding to represent the word information, we use a scalar $\alpha$ for each word to indicate how likely the word is superficial. We multiply the word embedding by $\alpha$ for each word. Note that one word in different positions should have a unique $\alpha$, since we assume they correspond to the same symbol and thereby whether they are superficial are identical.

**Correspondence Information Representation** In theorem 2, although we can replace each symbol to a new symbol, the symbols should be replaced accordingly. So for the superficial non-logical symbols, their correspondence information affects the inference. This can be easily illustrated from the perspective of NLI in figure 1a. If we discard the superficial symbols but reserve their correspondence information, we will get $s_1'$ and $s_2'$, from which their contradiction can be still inferred. But if we discard both the superficial symbols and their correspondence information to get $s_1''$ and $s_2''$, their relation is infeasible to infer. In order to represent the correspondence information, we use a graph neural network which connects the same words in different positions of the word pairs. Thus the correspondence information is able to propagate through these positions.

### 4.4 NEURAL NETWORK IMPLEMENTATION

**Architecture** Our proposed neural network consists of three major modules, which is shown in figure 3. The first module is the superficial information projection module, which is motivated by the word information discard in section 4.3. For each word $w_i$, we compute its superficial factor $\alpha_i$, which is a scalar indicating how superficial the word is. $\alpha_i = 1$ means the word corresponds to non-logical symbols that we want to keep the information during inference, or the word corresponds to a logical symbol. $\alpha_w = 0$ means the word is totally useless. The embedding of each word is multiplied by the $\alpha_i$.

The second module is a standard NLI model. We can use arbitrary NLI models (e.g. ESIM Chen et al. (2017), MwAN Tan et al. (2018)) as this module. The output of this model is a sequence of embeddings, indicating the states of the words.

The third module represents the correspondence information in section 4.3. We need to keep the correspondence of the superficial symbols via a graph neural network.

**Superficial information projection** To discard the words with superficial information, we multiply the embedding of each word by its superficial factor $\alpha$. More specifically, the embedding of a word $w_i$ is computed by:

$$e = \alpha_i E w_i \tag{7}$$

, where $w_i$ is in the one-hot representation, $E$ is the embedding matrix.

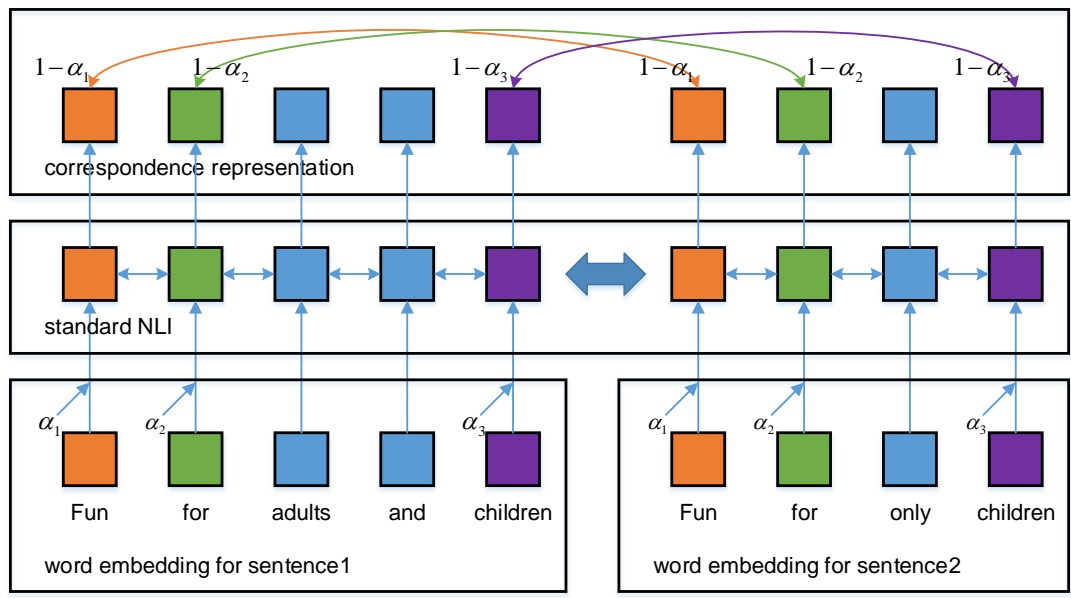

Figure 3: Architecture of the proposed neural network.

Note that $\alpha_i$ is the same for one word in different positions of the sentence pair. To achieve this, we simply use a single perceptron layer over the embeddings to compute such $\alpha$.

$$\alpha_i = \sigma(M[Ew_i; t_i] + b) \tag{8}$$

, where $M$ is the parameter matrix for $\alpha$, $[;]$ denotes the concatenation operation, $t_i$ denotes whether $w_i$ is overlapped in the sentence pair ($t_i = 1$) or not ($t_i = 0$).

**Correspondence representation** To represent the cross-sentence correspondence information, we use a graph neural network. For the same word which occurs in different positions in the sentence pair, we use an edge between all position pairs to represent the correspondence information. Intuitively, for words that are superficial, we only need to retain their correspondence information, and vice versa. As $\alpha_i$ denotes whether the information should be retained, we set the weight of the edge to $1 - \alpha_i$ for word $w_i$. More formally, we denote the states at time as $S^T \in \mathbb{R}^{n \times d}$, where $n$ is the total length of the sentence pair and $d$ is the dimension of the hidden states. By following the graph neural network in Kipf and Welling (2016), we update $S^T$ by:

$$S^T = \sigma(AS^{T-1}W^T) \tag{9}$$

, where $W^T$ is the parameter matrix, $S^0$ is the output of the standard NLI module, and $A \in \mathbb{R}^{n \times n}$ is the adjacency matrix to represent such correspondence:

$$A_{i,j} = \begin{cases} \alpha_i & \text{if } i = j \\ \theta(1 - \alpha_i) & \text{if } i \neq j, w_i = w_j \\ 0 & \text{otherwise.} \end{cases} \tag{10}$$

, where $\theta$ is used to make the sum of each row in $A$ equals to 1. Figure 3 show how we connect the words "fun", "for", and "children" in different positions in the sentence pair. By using the edges, even if the model discards the semantics of "children", it is able to represent that the word is behind "and" in the first sentence, and behind "only" in the second sentence. Therefore we retain the correspondence information by the graph neural network.

## 5 EXPERIMENTS

### 5.1 SETUP

**Datasets** We use the datasets including MNLI (Williams et al., 2018), SNLI (Bowman et al., 2015), QNLI (Wang et al., 2018), DNLI (Welleck et al., 2018), RTE (Dagan et al., 2005), MRPC (Dolan and Brockett, 2005), and SciTail (Khot et al., 2018). More details are shown in appendix D.

**Competitors** Since our proposed framework can use different NLI models as the second module, we use standard NLI models for both comparison and for NLI module. These models include BiLSTM, ESIM (Chen et al., 2017), MwAN (Tan et al., 2018), and CAFE (Tay et al., 2018). We compare with HEX (Wang et al., 2019), which projects superficial statistics out. We also compare with the pre-training model Elmo (Peters et al., 2018), Roberta (Liu et al., 2019), which achieves state-of-the-art results in NLI. More details of the experimental setup are shown in appendix D and appendix E.

## 5.2 SINGLE DOMAIN EVALUATION

**Effectiveness** We evaluate the effectiveness of our proposed approaches in the single domain setting. The training and test data are from the same domain. Table 2 shows the performances of different models. $Ours+A$ denotes applying algorithm $A$ as the standard NLI module in our proposed neural network. Our proposed method constantly outperforms the original model by a large margin.

| Model | MRPC | RTE | QNLI | SciTail | SNLI | MNLI | DNLI | DNLI(gold) | Avg. |
|---|---|---|---|---|---|---|---|---|---|
| HEX | 73.6/82.8 | 53.1 | 49.6 | 84.3 | 52.8 | 60.6/60.9 | 69.5 | 70.8 | 65.8 |
| BiLSTM | 69.7/80.5 | 54.7 | 74.0 | 77.0 | 82.5 | 68.8/69 | 86.7 | 91.7 | 75.5 |
| BiLSTM+ours | 77.6/84.5 | 58.5 | 80.3 | 83.8 | 85.5 | 75.2/74.2 | 87.0 | 91.4 | **79.8(+4.3)** |
| ESIM | 68.7/80.8 | 53.4 | 80.9 | 82.8 | 88.1 | 77.4/76.7 | 87.9 | 92.8 | 79.0 |
| ESIM+ours | 76.9/84.1 | 57.6 | 80.8 | 84.1 | 88.3 | 78.5/77.5 | 88.7 | 93.2 | **81.0(+2.0)** |
| MwAN | 68.8/80.7 | 51.9 | 69.4 | 71.2 | 85.2 | 74.1/73.3 | 86.0 | 90.3 | 75.1 |
| MwAN+ours | 76.7/83.6 | 59.9 | 81.9 | 84.2 | 82.6 | 73/73 | 85.3 | 88.9 | **78.9(+3.8)** |
| CAFE | 69.1/80.6 | 53.4 | 82.2 | 81.2 | 86.8 | 76.3/76 | 88.1 | 92.8 | 78.7 |
| CAFE+ours | 76.5/83.8 | 58.4 | 83.6 | 85.6 | 86.4 | 75.2/74.7 | 89.0 | 93.3 | **80.7(+2.0)** |

Table 2: Performance over single domain NLI and single domain PI (MRPC). For MRPC, we report the accuracy and f1-score. For MNLI, we report the accuracy on both matched and mismatched test sets. For the rest datasets, we report the accuracy.

**Ablations** We evaluate the effectiveness of the two inductive biases in section 4.3, i.e., word information discard and correspondence information representation. We use an ablation study in Table 3 to evaluate them. Here $-word$ means no word discard (i.e. $\alpha$ only works in the correspondence representation module). $-correspond$ means no correspondence representation module. From the results, both inductive biases improve the effectiveness. The word information discard is more crucial.

| Model | MRPC | RTE | QNLI | SciTail | SNLI | MNLI | DNLI | DNLI(gold) | Avg. |
|---|---|---|---|---|---|---|---|---|---|
| BiLSTM+ours | 77.6/84.5 | 58.5 | 80.3 | 83.8 | 85.5 | 75.2/74.2 | 87.0 | 91.4 | 79.8 |
| -correspond | 77/84.1 | 60.1 | 79.9 | 77.1 | 86.1 | 75/74.3 | 86.7 | 91.2 | 79.2(-0.6) |
| -word | 74.3/82.5 | 50.9 | 80.2 | 82.4 | 84.2 | 72.5/71.8 | 85.9 | 90.3 | 77.5(-2.3) |
| ESIM+ours | 76.9/84.1 | 57.6 | 80.8 | 84.1 | 88.3 | 78.5/77.5 | 88.7 | 93.2 | 81.0 |
| -correspond | 75.1/83.3 | 58.6 | 81.5 | 83.8 | 88.1 | 78.6/77 | 88.6 | 93.3 | 80.8(-0.2) |
| -word | 69.4/80.1 | 55.6 | 80.2 | 83.6 | 88.2 | 77.9/76.9 | 88.7 | 93.2 | 79.4(-1.6) |

Table 3: Ablation over single domains.

## 5.3 RESULTS OVER PRE-TRAINING MODELS

State-of-the-art NLI results are from the fine-tuning of pre-training models. We use Elmo (Peters et al., 2018) and Roberta Liu et al. (2019), a recent pre-training model, as the word embeddings module in our architecture Liu et al. (2019). We use the pooling layer in ESIM for final classification. The results are shown in Table 4. While our proposed method outperforms the original ESIM+ELMO by a large margin, the accuracies are slightly improved for Roberta. This makes sense because Roberta already reached a very high accuracy.

## 5.4 EVALUATION FOR UNSEEN DOMAINS

We evaluate the robustness of our approaches in unseen domains. We choose one dataset as the source domain for training, and another dataset as the target unseen domain for testing. The model is only trained by the training data in the source domain.

| Model | MRPC | RTE | QNLI | SciTail | SNLI | MNLI | DNLI | DNLI(gold) | Avg. |
|---|---|---|---|---|---|---|---|---|---|
| Elmo+ESIM | 70.8/81.4 | 54.1 | 81.3 | 81.8 | 88.4 | 79.7/78.5 | 88.8 | **93.7** | 79.9 |
| Elmo+ESIM+ours | **80.0/85.8** | **60.8** | **82.5** | **86.3** | **88.7** | **79.8/79.2** | **89.2** | 93.5 | **82.6(+2.7)** |
| Roberta | 88.2/91.4 | 72.1 | 92.6 | **93.6** | 91.0 | 87.2/86.8 | 91.2 | **95.9** | 89.0 |
| Roberta+ours | **88.6/91.6** | **73.1** | **92.8** | 93.5 | **91.4** | **87.3/86.7** | **91.6** | **95.9** | **89.3(+0.3)** |

Table 4: Results over pre-training models.

| | NLI (3 classes) | | | | | | | | NLI(2 classes) | | |
|---|---|---|---|---|---|---|---|---|---|---|---|
| Source | DNLI | DNLI | MNLI | MNLI | | SNLI | | SNLI | RTE | SciTail | AVG. |
| Target | SNLI | MNLI | SNLI | DNLI | Gold | DNLI | Gold | MNLI | SciTail | RTE | |
| HEX | 33.3 | 36.9/36.5 | 52.8 | 49.6 | 50.9 | 34.4 | 50.9 | 38.0/38.5 | 52.9 | 53.4 | 44.0 |
| BiLSTM | 37.0 | 38.5/38.2 | 54.5 | 46.5 | 48.9 | 39.4 | 40.4 | 54.3/56.1 | 47.3 | 54.1 | 46.3 |
| BiLSTM+ours | 36.4 | 37.4/37.9 | 64.1 | 56.2 | 58.7 | 46.9 | 48.3 | 60.7/60.2 | 63.5 | 56.7 | **52.3(+6.0)** |
| ESIM | 36.7 | 37.2/37.5 | 68.1 | 61.4 | 64.8 | 47.5 | 48.8 | 62.9/62.6 | 55.8 | 55.6 | 53.2 |
| ESIM+ours | 37.7 | 38.5/39.6 | 69.2 | 62.2 | 65.3 | 48.8 | 49.9 | 63.4/63.5 | 57.3 | 58.4 | **54.5(+1.3)** |
| MwAN | 38.0 | 38.4/38.2 | 63.6 | 55.5 | 58.7 | 39.7 | 40.3 | 58.2/58.9 | 55.1 | 49.7 | 49.5 |
| MwAN+ours | 36.9 | 38.9/39.9 | 62.0 | 57.4 | 60.3 | 48.1 | 49.3 | 59.3/59.3 | 54.2 | 58.6 | **52.0(+2.5)** |
| CAFE | 37.9 | 38.5/39.2 | 67.5 | 60.1 | 63.5 | 48.2 | 49.7 | 62.1/61.4 | 41.4 | 56.1 | 52.1 |
| CAFE+ours | 38.0 | 37.6/38.4 | 67.8 | 59.6 | 63.1 | 48.0 | 49.4 | 62.3/62.2 | 60.1 | 56.3 | **53.6(+1.5)** |

Table 5: Performance in unseen domains. "Gold" denotes the gold-standard test set of DNLI.

Table 5 shows the performance of different models. From the results, we see that by using our proposed method, the accuracy improves significantly.

## 5.5 VISUALIZATION OF THE PROJECTION

We visualize the $\alpha$ to deeply analyze its performance in Figure 4. Each grid of a word represent its $\alpha$. Our approach successfully projects superficial words out. For example, in figure 4a, the words "women" and "bar" are mostly discarded, while both words do not affect the inference. The same intuitive discarding happens in the words "man" and "shirt" in figure 4b. We also visualize and analyze the attention mechanism in appendix G.

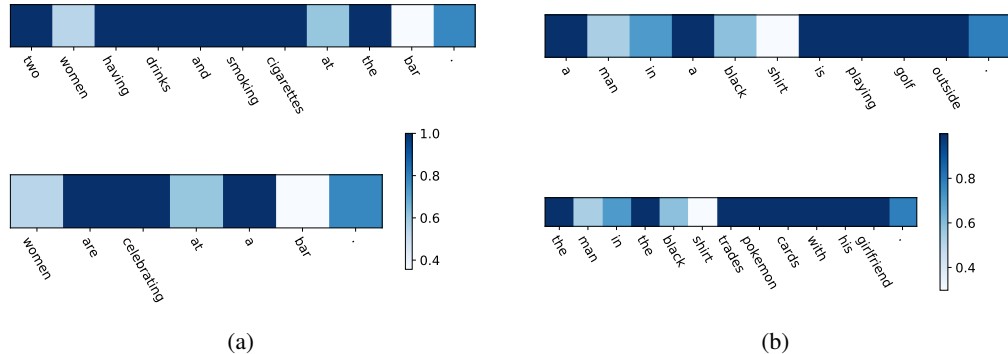

(a)  (b)

Figure 4: $\alpha$ for two sentence pairs.

## 6 CONCLUSION

In this paper, we study the problem of projecting superficial information out for NLI. The projection prevents models from overfitting and makes them more robust. Specially, we explain the superficial information from the perspective of FOL, and project them out in a neural network-based architecture. We conduct extensive experiments to verify the effectiveness of our proposed approach. The results verify that our proposed approaches increase the baselines by a large margin.

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

## A    FORMAL PROBLEM SETUP FOR THE SINGLE DOMAIN AND THE UNSEEN DOMAIN

**Single domain** In the single domain setting, the training data and the test data have the same distribution $P_X, y$. More formally, given training data $Train = \{x_n, y_n \sim P_{X,y}\}_{n=1}^N$, the goal is to

predict the labels of the test data $Test = \{x_m, y_m \sim P_{X,y}\}_{m=1}^M$. Each $x_i$ consists of two sentences $s_i^1, s_i^2$. For example, in NLI, each $y_i \in \{neutral, contradiction, entailment\}$ indicates the relation between $s_i^1$ and $s_i^2$. Each sentence is a sequence of words.

**Unseen domains** To evaluate the generality of the model after discarding superficial information, we also consider its effectiveness in an unseen domain. More formally, suppose the source domain and the target domain have distribution $P_X^{(s)}, y$ and $P_X^{(t)}, y$, respectively. We evaluate the model that is trained on the source domain $Train^{(s)} = \{x_n^{(s)}, y_n^{(s)} \sim P_{X,y}^{(s)}\}_{n=1}^N$ and is tested on the target domain $Test^{(t)} = \{x_m^{(t)}, y_m^{(t)} \sim P_{X,y}^{(t)}\}_{m=1}^M$. Note that in the unseen domain NLI, $P_X^{(t)}, y$ and $Test^{(t)}$ are unknown during training. This setting is more challenging than traditional domain-adaptation (Ajakan et al., 2014; Cui et al., 2019) and domain generalization (Muandet et al., 2013) from the perspective that the test domain is unknown during training.

## B    THE SYNTAX OF FOL

| Syntax of the atoms in FOL | Symbol type |
|---|---|
| $Connective \rightarrow \vee \mid \wedge \mid \Rightarrow$ | logical |
| $Quantifier \rightarrow \forall \mid \exists$ | logical |
| $Constant \rightarrow A \mid VegetarianFood \mid \cdots$ | non-logical |
| $Varaible \rightarrow x \mid y \mid \cdots$ | non-logical |
| $Predicate \rightarrow Serves \mid Near \mid \cdots$ | non-logical |
| $Function \rightarrow LocationOf \mid CuisineOf \mid \cdots$ | non-logical |

Table 6: The syntax of FOL, specified in Backus-Naur form (Russell and Norvig, 1995).

## C    PROOF OF THEOREM 1

*Proof.* For a non-logical symbol $ns$, since $\forall CS$,

$$CS \wedge FOL(s_1) \models FOL(s_2) \tag{11}$$

is equivalent to

$$CS \wedge FOL'(s_1) \models FOL'(s_2) \tag{12}$$

For $CS = True$, $CS \wedge FOL(s_1) = FOL(s_1)$, $CS \wedge FOL'(s_1) = FOL'(s_1)$. Thus for a non-logical symbol $ns$, we have

$$FOL(s_1) \models FOL(s_2) \tag{13}$$

is equivalent to

$$FOL'(s_1) \models FOL'(s_2) \tag{14}$$

$\square$

## D    EXPERIMENTAL SETTINGS AND DATASETS

All the experiments run over a computer with Intel Core i7 4.0GHz CPU, 32GB RAM, and a GeForce GTX 1080 Ti GPU. For SNLI, we remove the *"the other"* category to make its labels comparable with MNLI. We evaluate the accuracy and f1-score for MRPC, since its labels are imbalanced. We list the statistics of the datasets in Table 7.

## E    NEURAL NETWORK DETAILS

We use a pooling layer over our proposed architecture for sentence pair classification. For ESIM, MwAN and CAFE, we use the pooling layer in their original papers. For BiLSTM, we follow (Wang et al., 2018). We use a max pooling layer to produce the vectors $u$, $v$ of each sentence, and pass $[u; v; |u - v|; u * v]$ to an MLP classifier which has a hidden layer with $tanh$ activation. We apply $softmax$ over the output layer.

|       | #Train | #Dev       | #Test       | Avg.L | Vocab. |
|-------|--------|------------|-------------|-------|--------|
| MNLI  | 392702 | 9815/9832  | 9796/9847   | 17.0  | 95871  |
| SNLI  | 549367 | 9842       | 9824        | 11.2  | 40257  |
| MRPC  | 4076   | -          | 1725        | 22.3  | 18765  |
| QNLI  | 104743 | 5463       | 5463        | 21.2  | 89686  |
| RTE   | 2490   | 277        | 3000        | 29.5  | 26751  |
| DNLI  | 310110 | 16500      | 16500/12376 | 9.4   | 15568  |
| SciTail| 23596 | 1304       | 2126        | 16.4  | 28169  |

Table 7: Statistics of datasets. Avg.L denotes the average length of each review. Vocab. denotes the vocabulary size. We show the size of the matched/mismatched test sets for MNLI.

For Roberta, we split the outputs of Roberta into $v_a$, $v_b$ according to the $\langle SEP \rangle$ separator. Then we use the pooling layer in ESIM. We use our proposed method and its ablations, and choose the best model w.r.t. the developing dataset.

Hex (Wang et al., 2019) relies on a textural model to generate superficial information, and a raw model to generate all information. We use a two-layer BiLSTM over the overlapping words to generate the superficial information. And we use another BiLSTM over the raw sentence to generate all information.

**Hyper-parameters** For BiLSTM, the dimension of the hidden states is set to 300. For other models, we use the dimension of the hidden states as their original papers. For ESIM, CAFE, and MwAN, their dimensions are set to 300, 300, 75, respectively. We use the AMSGrad (Reddi et al., 2018) optimizer except Roberta, in which we use AdamW (Loshchilov and Hutter, 2019). We use 300d GloVe vectors (Pennington et al., 2014) as the initialization for the word embedding except Roberta. For the $T$-layer graph neural network, to achieve the best performance, we set $T = 3$ for BiLSTM, ESIM and Roberta, $T = 2$ for MwAN, and $T = 1$ for CAFE.

## F ABLATIONS OVER UNSEEN DOMAINS

We show the ablations over unseen domains in table 8.

| Source
Target | DNLI
SNLI | DNLI
MNLI | MNLI
SNLI | MNLI
DNLI | Gold | SNLI
DNLI | Gold | SNLI
MNLI | RTE
SciTail | SciTail
RTE | AVG. |
|------------------|-----------|-----------|-----------|------|------|------|------|-----------|---------|---------|-----------|
|                  | NLI (3 classes) | | | | | | | | NLI(2 classes) | | |
| BiLSTM+ours | 36.4 | 37.4/37.9 | 64.1 | 56.2 | 58.7 | 46.9 | 48.3 | 60.7/60.2 | 63.5 | 56.7 | 52.3 |
| -correspond | 36.0 | 38.2/38.1 | 65.2 | 57.2 | 60.2 | 46.3 | 47.9 | 60.7/60.3 | 60.9 | 55.4 | 52.2(-0.1) |
| -word | 37.8 | 40.1/39.3 | 61.2 | 50.1 | 52.3 | 40.6 | 40.7 | 57.6/57.1 | 48.5 | 53.5 | 48.2(-4.1) |
| ESIM+ours | 37.7 | 38.5/39.6 | 69.2 | 62.2 | 65.3 | 48.8 | 49.9 | 63.4/63.5 | 57.3 | 58.4 | 54.5 |
| -correspond | 37.0 | 38.2/38.5 | 68.8 | 61.0 | 64.1 | 50.0 | 51.5 | 64.9/64.0 | 56.3 | 59.0 | 54.4(-0.1) |
| -word | 36.0 | 37.3/38.1 | 67.3 | 59.8 | 62.7 | 46.7 | 47.7 | 62.5/62.5 | 49.7 | 55.6 | 52.2(-2.3) |

Table 8: Ablations over unseen domains. For MNLI, we report the accuracy on both matched and mismatched test sets. "Gold" denotes the gold-standard test set of DNLI.

## G EFFECT OF THE SUPERFICIAL INFORMATION IN ATTENTION

We further investigate what is projected and what is reserved from the attention perspective. Figure 5 show the attention matrix w/o discarding superficial information in ESIM. Clearly, the attention matrix after discarding superficial information is more intuitive. It concentrates on "usual" and "slightly lower", which imply the contradiction relation. In contrast, the matrix of the standard ESIM focus on the repeated words (e.g. "Toronto", "stock"), which are not critical for the inference.

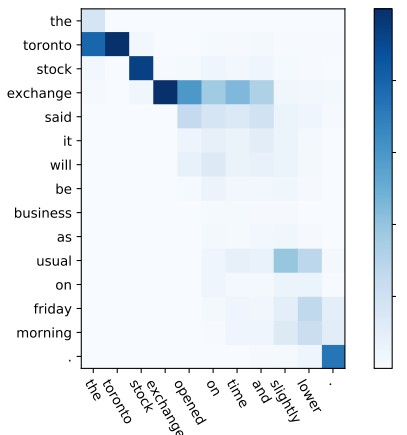

(a) Attention matrix in standard ESIM.

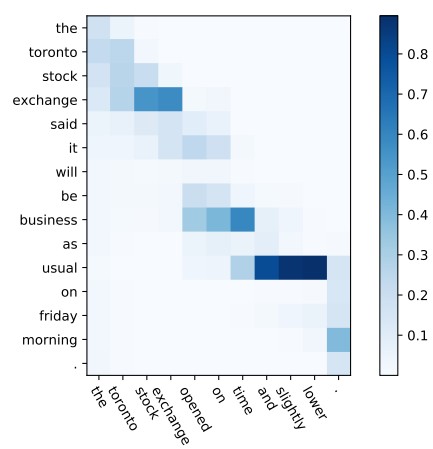

(b) Attention matrix in our method.

Figure 5: Attention matrix visualization. The x-axis and the y-axis denote words in the two sentences.

