# OpenReview forum: "Robust Natural Language Representation Learning for Natural Language Inference by Projecting Superficial Words out"
_ICLR.cc/2020/Conference — Reject_

### Official Review · AnonReviewer1 · 2019-10-23
**Official Blind Review #1**

**Rating:** 1

**Review:**

This paper uses first order logic (FOL) to help reduce so-called “superficial” information/semantics that is less relevant to the judgement of natural language inference relations. The submission misses the major literature of and comparison to previous work that uses FOL for natural language inference (aka. RTE), for example, [Bos and Markert ‘05], [Beltagy et al. ‘13], [Abzianidze ‘17], among others, as well as work based on natural logic, e.g., [MacCartney ‘09], which operates directly on parsed sentences. The submission contains little contribution with regard to the exiting work. Key concepts such as “superficial semantics” is vague and not well defined. I do not recommend it for the conference.

Bos and Markert ‘05, Recognising Textual Entailment with Robust Logical Inference.
Beltagy et al. ‘13, Montague Meets Markov: Deep Semantics with Probabilistic Logical Form.
Abzianidze ’17, A Natural Proof System for Natural Language.
MacCartney ’09, natural language inference (PhD thesis)


**Experience Assessment:**

I have published in this field for several years.

**Review Assessment: Checking Correctness Of Derivations And Theory:**

I assessed the sensibility of the derivations and theory.

**Review Assessment: Checking Correctness Of Experiments:**

I did not assess the experiments.

**Review Assessment: Thoroughness In Paper Reading:**

I read the paper at least twice and used my best judgement in assessing the paper.

---

### Official Review · AnonReviewer2 · 2019-10-24
**Official Blind Review #2**

**Rating:** 3

**Review:**

This paper tried to reduce superficial information in natural language inference (NLI) to prevent overfitting. It utilized the first order logic to explain what is superficial information.
Then, it introduced a superficial factors in the existing neural networks. Furthermore, they introduce a graph neural network (GNN) to model relation between premise and hypothesis.
It was evaluated on a bunch of NLI benchmarks including SNLI, SciTail, MNLI etc, showing the effectiveness of the proposed model.

This paper is well motivated and the ideas are interesting. However, there are a few concerns detailed as follows:

1. Do these methods only work on small tasks? For example, the big improvement only appears in small tasks such as MRPC and RTE. However, the proposed method experiences performance decreases on large tasks such as SNLI and MNLI. E.g., in CAFE settings, the proposed approach got  75.2/74.7 (matched/mismatched) vs 76.3/76 (baselines). The similar observations are found in MwAN settings on MNLI and SNLI. I’d like to see some discussion in the paper on this.

2. What common logic patterns did the model learn when removing all the superficials? For different relations, e.g., entailment vs contradiction, are these patterns different? This may help the reader understand whether the model really filtered these information.

3. It requires more discussions on Table 5. E.g., in NLI (2 classes), the random guess should be 50%. But the model performance was 41.4% on CAFE when transferring from RTE to SciTail, which is even worse than random guess. In contrary, when transferring from SciTail to RTE, the model performance was 56.1%, which seems reasonable. I believe more analysis and discussions are required to understand this model.


**Experience Assessment:**

I have published one or two papers in this area.

**Review Assessment: Checking Correctness Of Derivations And Theory:**

I assessed the sensibility of the derivations and theory.

**Review Assessment: Checking Correctness Of Experiments:**

I carefully checked the experiments.

**Review Assessment: Thoroughness In Paper Reading:**

I read the paper at least twice and used my best judgement in assessing the paper.

---

### Official Review · AnonReviewer3 · 2019-10-24
**Official Blind Review #3**

**Rating:** 1

**Review:**

This paper presents an approach to treat natural language inference using first-order logic, and to infuse neural NLI models with logical information to be more robust at inference. However, the paper does not contain a single reference to the computational semantics literature, where logical approaches towards semantics were the dominant trend for many years (see e.g. [1, 2]). Indeed, 'neuralising' first order logic has been an active area of recent research ([3] or indeed much of the recent work coming from Sebastian Riedel's group). This is a glaring oversight.

The paper starts by introducing background on first-order logic, and then gives a definition of a 'superficial' predicate, namely one whose extension is not necessary to prove an implication for any collection of background facts. However, by extension, this makes s_1 -> s_2 a tautology, which is the 'true' notion that the authors are looking for. Indeed, if |- (s_1 -> s_2), then for any collection of formulae \Delta then \Delta |- (s_1 -> s_2) (by monotonicity of entailment) and clearly if for any \Delta we have \Delta |- (s_1 -> s_2), we can take \Delta to be the empty set. Finally, the authors show that tautologies are still tautologies under change of predicates (i.e. if we only require logical rules to prove one statement from another, then the extensions of predicates in those statements do not matter).

The authors then use this to motivate two extensions to inference models. One is to 'drop out' word information, and the other is to treat different occurrences of the same word as reflecting the same underlying predicate. The first somewhat transparently forces the model to care less about the exact meaning (i.e. extension in the logical world) of words (indeed, word vectors have been shown to capture extensional information [4, 5]), and so may force the inference model to learn more 'logical' inference rules. Further, the word dropout calculation includes whether the word is in both sentences, which is a strong signal that its extension may not be necessary. However, the second only forces the intuition that different mentions of the same word are likely to be coreferent, which is a weak assumption that models may already pick up. Indeed, it is noticeable that this component seems to be less necessary in the authors' ablation study.

In summary, while I am sympathetic to the aim of grounding neural models in explicit notions of semantics, this paper shows such a lack of awareness of previous literature that I cannot recommend acceptance.

[1] The Meaning Factory: Formal Semantics for Recognizing Textual Entailment and Determining Semantic Similarity, Bjerva et al. 2014
[2] Natural Logic for Textual Inference, MacCartney and Manning 2009
[3] End-to-end Differentiable Proving, Rocktaschel and Riedel 2017
[4] Building a shared world: mapping distributional to model-theoretic semantic spaces, Vecchi and Herbelot 2015
[5] Deriving Boolean structures from distributional vectors, Kreuzewski et al 2015

**Experience Assessment:**

I have read many papers in this area.

**Review Assessment: Checking Correctness Of Derivations And Theory:**

I carefully checked the derivations and theory.

**Review Assessment: Checking Correctness Of Experiments:**

I carefully checked the experiments.

**Review Assessment: Thoroughness In Paper Reading:**

I read the paper thoroughly.

---

### Decision · Program_Chairs · 2019-12-19

**Decision:**

Reject

**Comment:**

This paper proposes using first order logic to rule out superficial information for improved natural language inference. While the topic is of interest, reviewers find that the paper misses much of the previous literature on semantics which is highly relevant.

I thank the authors for submitting this paper to ICLR. Please take the reviewers' comments, especially recommended references, to improve the paper for future submission.